# Assessing Organizational Cultural Responsiveness among Refugee-Servicing Domestic Violence Agencies

**Jessica L. Lucero [1],*, Kristina M. Scharp [2]  and Tanni Hernandez [3]**

[1]  Department of Sociology, Social Work, and Anthropology, Utah State University, Logan, UT 84322, USA
[2]  Department of Communication, University of Washington, Seattle, WA 98195, USA; kscharp@uw.edu
[3]  Touchstone Therapy Center, St. George, UT 84790, USA; t.a.hernandez@hotmail.com
*  Correspondence: Jessica.lucero@usu.edu

**Abstract:** Refugee community members who have experienced domestic violence in the U.S. face complex challenges in seeking help which may ultimately impact their ability to leave violent relationships. When domestic violence organizations are not prepared to serve them in culturally responsive ways, these challenges are exacerbated. This study surveyed 70 executive directors of domestic violence agencies in U.S. resettlement cities about the extent to which their organization's practices reflected cultural responsiveness in serving refugee populations. The results showed promising indicators of organizational cultural responsiveness but uncovered numerous areas for growth. In particular, the study results underscore the need for organizations to improve their language supports and take active steps to outreach to, hire, and engage refugee communities in order to better serve them. This paper makes recommendations for how DV agencies can be more culturally responsive as they support refugee individuals who are seeking safety from violent relationships.

**Keywords:** refugees; domestic violence; cultural competence; organizational cultural responsiveness; diverse populations

## 1. Introduction

Domestic violence (DV) is a serious, global public health issue with long-term negative impacts on individuals, families, and societies (World Health Organization 2017). Domestic violence does not discriminate by income, race/ethnicity, nationality, belief system, gender, age, ability, or any other identity group. Although DV does not appear to be any more prevalent among refugees than it is among the general population, refugees experience unique challenges that might exacerbate their violent experience, make it difficult to seek help, and ultimately, impact their ability to leave a violent relationship (Menjívar and Salcido 2002; Runner et al. 2009).

Since 1980, approximately 3 million refugees have been resettled in the U.S. (Pew Research Center 2019). In the last decade, refugees from Burma, Iraq, and Bhutan have been the largest groups resettled in the U.S.'s 212 resettlement cities and more recently, refugees from the Democratic Republic of Congo, Ukraine, Syria, Eritrea, and Somalia have resettled in larger numbers (Blizzard and Batalova 2019). Although the last few years have witnessed stark declines in refugee resettlement numbers in the U.S., the number of refugees globally is at a historic high (Pew Research Center 2019). Refugee communities should not be viewed as monolithic—each community has diverse sociocultural characteristics. Nevertheless, a unifying theme among newly resettled refugees is the difficulty they often experience in system navigation and help-seeking. For refugees affected by DV, challenges related to language, culture, migration and trauma histories, and gender roles all coalesce to make DV system

navigation more difficult and help-seeking less likely (Bhuyan et al. 2005; Crandall et al. 2005; Latta and Goodman 2005; Shiu-Thornton et al. 2005; Sullivan et al. 2005). For example, refugees may experience challenges communicating with the police or social service providers, navigating a complex legal system, or filling out any required paperwork along the way (Bhuyan et al. 2005; Crandall et al. 2005; Latta and Goodman 2005; Menjívar and Salcido 2002; Ortiz Hendricks 2009; Shiu-Thornton et al. 2005; Sullivan et al. 2005). Newly resettled refugees may not understand their rights in a U.S. context and may fear deportation of their partner if they disclose or report. Compounding these challenges, research shows that human service providers in the U.S. are often uncertain of how to effectively engage and intervene with the refugee community (Daniels and Belton 2015). If refugees face significant challenges in seeking help from formal DV service providers *and* service providers do not know how to support them, refugees who have been victimized by their partners may find it particularly difficult to leave a violent relationship and find safety. It is critical that DV service providers recognize the unique experiences and challenges that refugees face, develop and adapt programs that are culturally responsive, and actively outreach to refugee communities in order to more effectively serve this special population.

Given the growing refugee population in the U.S., there is a need among DV agencies to better understand how to effectively serve refugee survivors. Although there is a growing body of literature related to culturally responsive service provision in health contexts, there is very little empirical work that directly informs culturally responsive service provision in domestic violence contexts, specifically focused on refugee populations. Thus, the purpose of the current study is to assess culturally responsive organizational practices among domestic violence agencies in U.S. resettlement cities in an attempt to gain a baseline understanding of organizational cultural responsiveness and inform future service provision and research.

## 2. Background

### 2.1. Needs of Refugee and Immigrant Survivors

A small number of qualitative studies have explored the experiences, concerns, and greatest needs related to domestic violence among refugee communities, but a larger literature explores the same among immigrant communities in the U.S. The term refugee is often used interchangeably with the terms migrant or undocumented immigrant; however, the legal definition of refugee is qualitatively different from that of undocumented immigrant. Although refugees and undocumented immigrants experience similar traumatic trajectories brought on by persecution, war, or violence, in resettlement countries, refugees have legal status whereas undocumented immigrants do not. It is essential to understand the implications of this important difference and, even more, recognize that groups within either of these categories are not culturally monolithic. That said, the literature demonstrates considerable overlap in risk and contextual factors for refugees and immigrants and thus, we review the literature with this in mind.

There is a robust body of literature that identifies language as one of the primary areas of concern in DV situations as it affects both their experience with DV and their ability to seek help (Bhuyan et al. 2005; Crandall et al. 2005; Latta and Goodman 2005; Menjívar and Salcido 2002; Shiu-Thornton et al. 2005; Sullivan et al. 2005). Some abusers use language as a means of control, discouraging or preventing women from learning English and thus, further isolating them from potential sources of help (Bhuyan et al. 2005; Sullivan et al. 2005; Runner et al. 2009). Additionally, women who are not fluent in English experience barriers at every turn when they seek help for their abusive situation. They may have trouble communicating with the police or social service providers, navigating a complex legal system, or filling out required paperwork (Bhuyan et al. 2005; Crandall et al. 2005; Latta and Goodman 2005; Menjívar and Salcido 2002; Ortiz Hendricks 2009; Shiu-Thornton et al. 2005; Sullivan et al. 2005). Combined with cultural and societal views of DV in the refugee survivor's country of origin, these language issues help to explain why many refugee

and immigrant survivors lack awareness of domestic violence laws and available services, as well as their challenges in seeking help (Bhuyan et al. 2005; Crandall et al. 2005; Latta and Goodman 2005; Menjívar and Salcido 2002; Ortiz Hendricks 2009; Shiu-Thornton et al. 2005; Sullivan et al. 2005; Runner et al. 2009).

Research suggests that these challenges can be mitigated by culturally responsive practices (Bhuyan et al. 2005; Crandall et al. 2005; Latta and Goodman 2005; Menjívar and Salcido 2002; Ortiz Hendricks 2009; Shiu-Thornton et al. 2005; Sullivan et al. 2005; Runner et al. 2009). Refugees and immigrants recommend that agencies should provide translation and interpretation services and have someone familiar with the cultural background of their clients. Additionally, community outreach and education should be a priority, both to inform women about available services and to educate men about domestic violence laws in the U.S. Additionally, refugees and immigrants indicate that services to help them become more independent and adapt to living in the U.S would help support their exit from violent relationships. This could include assistance with obtaining a driver's license, finding employment, becoming more proficient in the English language, understanding the legal system, and accessing other services to help them and their families. This body of research is particularly important as it centers the perspectives of refugee and immigrant survivors and helps guide practical and theoretical work in the domain of culturally responsive service provision.

### 2.2. Cultural Responsiveness and Refugee Survivors

Cultural responsiveness is a critical factor and essential framework for providing relevant and effective services across multiple systems of care. Most health and social service systems are not inherently culturally responsive; however, evidence shows that when interventions within these systems are culturally adapted, they work better than non-adapted interventions and produce better outcomes for diverse populations (Bernal and Domenech-Rodriguez 2012). Beyond culturally adapted programs and interventions, Calzada and Suarez-Balcazar (2014) contend that culturally competent staff who administer programs from within a culturally responsive organizational climate are best positioned to serve the diverse needs of underrepresented groups. Their work reinforces Pyles and Kim (2005) study that highlighted the need for DV agencies specifically to adopt agency and system levels of cultural competence in order to best serve the needs of underserved DV survivors. This work is grounded in the idea that cultural competence is a multi-level phenomenon that must exist beyond the micro-level of service provider–survivor interaction.

There is an overwhelming literature base on cultural competence. For example, Shen (2015) review of cultural competence in the field of nursing identified 15 separate but overlapping models of cultural competence. Calzada and Suarez-Balcazar (2014) work integrates these disparate models and provides a heuristic way of understanding how cultural competence influences organizations. Extending Pyles and Kim (2005) and Hyde (2004) work on organizational cultural competence, they identify three levels (i.e., organizational, staff, and program) to characterize this framework and promote ongoing awareness, knowledge, and skill development for people serving the organization. Specific strategies for enacting cultural competence within these levels are based on substantial theoretical and empirical work across disciplines (see Georgetown University National Center for Cultural Competence 2020). According to Calzada and Suarez-Balcazar (2014), organizational cultural responsiveness helps agencies and providers develop and deliver more culturally appropriate services to diverse clients and can make those clients feel more comfortable with the service provider. By contrast, clients may feel misunderstood or even discriminated against when services are not culturally responsive, and this can discourage individuals and even entire communities from seeking needed services. Thus, cultural responsiveness is of special concern when working with refugee survivors of domestic violence.

### 2.2.1. Level 1: Culturally Competent Organizational Climate

Supported by the cognitive and behavioral cultural competence of staff, organizational leaders often begin by creating a mission and vision statement that embraces diverse and multicultural

practices. These mission and vision statements drive corresponding action plans to ensure organizational accountability and might include a variety of practices including cultural competence training, recruitment/promotion/retention of diverse staff, mentoring, translation/interpretation services, coordination of health/social workers, and inclusion of family members. Being culturally competent at an organizational level might also include policies and practices that facilitate the support of outreach and information dissemination in ways that are transparent and inclusive (Calzada and Suarez-Balcazar 2014). In the context of the present study, the organization's relationship with refugee communities is of special consideration, and agencies should collaborate with community advisers to examine issues, problem solve solutions, and evaluate the cultural fit of the strategies being implemented (Dana et al. 1992; Ortiz Hendricks 2009; Runner et al. 2009).

### 2.2.2. Level 2: Culturally Competent Staff and Service Providers

Just as staff can support a culturally competent organizational environment, so too can the environment foster increased competence among staff members. At this level, people are asked to think about how the organization's mission and vision interplays with their own personal beliefs, values, and attitudes. This look inwards might include comparing and contrasting practices of different cultural groups, taking an inventory of one's own explicit and implicit biases, and exploring one's capacity to serve culturally diverse clients. The integration of the cognitive and behavioral dimensions might be reflected in actions such as active listening or the asking of questions to gain knowledge about how best to support diverse families' health and wellbeing goals (Purnell 2002). In sum, culturally competent staff are those who have taken the time to immerse themselves in the cultural norms, communication practices, and celebrations of the people for whom they serve (Calzada and Suarez-Balcazar 2014). These culturally responsive staff and service providers contribute to the overall cultural responsiveness of the organization itself because they are part of the organizational climate and have the potential to advocate for programs and services that better serve the specific population or group with whom they are working (Calzada and Suarez-Balcazar 2014).

### 2.2.3. Level 3: Culturally Adapted/Responsive Programs and Evaluation

Finally, the implementation and evaluation of adapted programs illustrate the cultural competence of a well-supported organization and staff. Specifically, adapted programs are those that take into consideration the language, patterns of behavior, and cultural values of diverse families (e.g., translation, incorporating cultural values, offering childcare; Calzada and Suarez-Balcazar 2014). Existing research supports the value of these adaptations and suggests they have more efficacy than non-adapted programs (Bernal and Domenech-Rodriguez 2012). Scholars, however, warn against adapting programs that focus on specific characteristics of an ethnic group, promoting instead adaptations made based on theory and existing evidence-based programs that have been previously successful (Calzada and Suarez-Balcazar 2014). In particular, the literature consistently identifies language services as being especially critical (Calzada and Suarez-Balcazar 2014; Dana et al. 1992; Ortiz Hendricks 2009; Purnell et al. 2011; Runner et al. 2009). This is consistent with the need for language services as identified by refugee domestic violence survivors. Runner et al. (2009) asserted that language is such a paramount issue that even if agencies only used their limited funds to provide communication services, it would be of great benefit to their refugee and immigrant clients. Because understanding the efficacy of adaptation has such practical implications (i.e., whether that adaptation will be used again or in other contexts), evaluation becomes essential. Specifically, these evaluations should hold programs to standards that reflect cultural competence using both qualitative and quantitative methods (Calzada and Suarez-Balcazar 2014).

### 2.3. Present Study

Previous research has outlined the unique risks and cultural considerations for refugees who have experienced domestic violence. Much of this research is qualitative and offers valuable insight

from refugee survivors. These insights point to the practices and policies that agencies and helping professionals could enact to better serve diverse communities. Although there is a growing body of literature that underscores the importance of culturally responsive practices, little to no research exists that estimates the extent to which domestic violence organizations in the U.S. are enacting culturally responsive practices in their service provision for refugees. Thus, with Calzada and Suarez-Balcazar's assessment tool in mind, the current study's purpose is to assess culturally responsive organizational practices among domestic violence agencies in U.S. resettlement cities. Specifically, we ask:

(1) To what extent are domestic violence agencies in resettlement cities across the U.S. enacting culturally responsive practices for refugee survivors;

(2) What agency characteristics are associated with self-reported culturally competent practices within domestic violence agencies?

## 3. Methods

### 3.1. Study Design

The present study represents a cross-sectional assessment of domestic violence organizations in U.S. resettlement cities. A comprehensive list of domestic violence organizations in every resettlement city in the U.S. was identified using the National Coalition Against Domestic Violence's (NCADV) state-by-state online search tool. According to the U.S. Department of State, there are 212 resettlement cities in the United States. Resettlement cities were the geographic target because agencies in these locales were more likely to have some familiarity with refugee communities. In total, we identified 301 domestic violence organizations in these 212 resettlement cities. Only agencies who operated shelter services were included in the sampling frame. Executive director emails were obtained via web searches and by calling organizations for confirmation. After obtaining Institutional Review Board approval, the online survey was sent to all 301 identified executive directors. Approximately 24 emails bounced, bringing the total sampling frame to 277 domestic violence organizations. In total, 78 individuals responded to the survey on behalf of their agency following 2 reminder emails over the course of 6 weeks. Of these 78 responses, 70 were valid and complete. Ultimately, we obtained a 28% response rate—low numbers likely due in part to the heavy workload of executive directors and the hesitance of these individuals to speak as experts on the survey topic. In the end, DV agencies from resettlement cities in 31 states responded to the survey, demonstrating a wide range of geographic diversity. It is important to note that the sampling frame represents DV agencies in resettlement cities only, and not a national sample of DV agencies more broadly.

### 3.2. Sample

The sampling unit for this study was the domestic violence organization; however, we also collected key information from the executive director of each organization. On average, the organizations surveyed have served their communities for 46 years ($SD = 25.1$), employ a median of 28 employees, and receive support from 48 volunteers at any given time. The organizations surveyed provide support for a median of 950 clients each year, with a median of 20 who reportedly come from refugee communities. Refer to Table 1 for additional agency characteristics.

**Table 1.** Agency Characteristics.

| Agency Characteristic | Median |
|---|---|
| Number of Paid Staff | 27.5 |
| Number of Volunteers | 47.5 |
| Number of Clients Served Annually | 950 |
| Number of Refugee Clients Served Annually | 20 |
| Annual Operating Budget | USD 1,600,000 |
| Annual Translation Services Budget | USD 1100 |

### 3.3. Instrumentation

The research team developed the survey collaboratively with expert consultants from the International Women's House (IWH) in Atlanta, Georgia. IWH has been operating for more than 20 years, providing a safe haven and supportive services for women and children who are victims of family violence, sexual abuse, and human trafficking and focusing specifically on serving immigrant and refugee women and children with cultural sensitivity. In addition, Calzada and Suarez-Balcazar (2014) model of cultural responsiveness was used to develop assessment questions that queried the executive directors of domestic violence organizations regarding their organization's culturally responsive practices across system levels. Upon Institutional Review Board approval, the survey was administered to executive directors via a Qualtrics online survey. Overall, the survey consisted of 77 questions, some of which were open-ended, and most of which were Likert scale questions related to the frequency with which various culturally responsive policies, practices, and approaches were enacted in the agency. Finally, the survey collected demographic and background information from the executive director and the agency itself.

The majority of survey questions that assessed agency cultural responsiveness fell into the following six domains: (1) agency values and practices (9 items, $\alpha = 0.903$); (2) agency staffing and programmatic practices (9 items, $\alpha = 0.899$); (3) interagency collaboration and outreach (6 items, $\alpha = 0.783$); (4) agency systems and processes for assessment (4 items, $\alpha = 0.846$); (5) program and shelter accommodations and adaptations (5 items, $\alpha = 0.688$); and (6) agency approaches to language concerns (9 items, $\alpha = 0.714$). These Likert scale questions were developed using Calzada and Suarez-Balcazar's assessment tool in consultation with IWH expert input. For bivariate analyses, single items in each domain were averaged to reflect a composite score. High levels of reliability indicate the appropriateness of treating these domains as larger constructs for correlation analyses. Specific items can be seen in Table 2 below.

**Table 2.** Culturally Responsive Organizational Practices Domains.

| My Agency . . . | M | SD |
|---|---|---|
| *Agency Values and Practices* (α = 0.903) | | |
| purchases resources (e.g., books, training materials, etc.) that create a culture of learning about the refugee community. | 2.61 | 0.931 |
| staff are knowledgeable of the cultural practices, values, and norms in the refugee community that may impact help-seeking among refugee victims of intimate partner violence. | 3.16 | 0.792 |
| actively outreaches to the refugee community to improve attitudes and social norms regarding intimate partner violence. | 2.82 | 0.962 |
| actively develops domestic violence leaders/advocates from within the refugee community. | 2.06 | 0.956 |
| forms collaborative partnerships with the refugee community. | 2.98 | 0.924 |
| works with consultants, such as culture specific healers or cultural brokers, who have in-depth knowledge of the refugee community. | 2.34 | 0.961 |
| engages in advocacy work within the refugee community. | 2.72 | 1.051 |
| depicts refugee clients in promotional materials. | 2.42 | 1.012 |
| displays decor and/or artwork from refugee communities in agency spaces. | 2.16 | 1.037 |
| *Agency Staffing and Programmatic Practices* (α = 0.899) | | |
| recruits bicultural and bilingual staff. | 3.43 | 0.816 |
| mentors bicultural and bilingual staff. | 3.23 | 0.973 |
| promotes bicultural and bilingual staff. | 3.31 | 0.903 |
| hires staff specifically from the refugee community to help with engagement and outreach. | 2.12 | 0.971 |
| encourages staff to engage in cultural immersion in the refugee community. | 2.45 | 1.1 |
| hosts multicultural events to promote cultural immersion in the refugee community. | 2.08 | 0.954 |
| creates opportunities for ongoing dialogue for staff to reflect on what is and what is not working when serving refugee individuals. | 2.81 | 1.065 |
| participates in interagency forums to reflect on what is and what is not working when serving refugee individuals. | 2.71 | 0.988 |
| provides formal, ongoing cultural competence training for staff. | 2.96 | 1.031 |
| *Inter-Agency Relationships and Outreach Programs* (α = 0.783) | | |
| actively works to build relationships with other DV service providers with expertise in issues that affect refugee clients. | 3.45 | 0.765 |
| actively outreaches to police to educate and develop coordinated care for the refugee community. | 2.46 | 1.091 |
| actively outreaches to medical providers to educate and develop coordinated care for the refugee community. | 2.19 | 1.045 |
| actively outreaches to faith leaders from the refugee community to improve their responses to intimate partner violence in their congregations. | 2.39 | 0.953 |
| actively partners with Refugee Services Offices (e.g., Catholic Charities, etc.) | 3.06 | 0.998 |

| My Agency . . . | *M* | *SD* |
|---|---|---|
| **Agency Systems and Processes for Assessment** (α = 0.846) | | |
| implements systems of accountability for cultural competence standards when working with refugee populations. | 2.42 | 1.048 |
| engages in assessment of cultural competence practices (e.g., collecting feedback from refugee clients). | 2.77 | 1.171 |
| engages in needs assessments of the refugee community. | 2.17 | 1.098 |
| uses culturally-relevant screening and assessment tools that have been adapted for the refugee community. | 2.25 | 1.082 |
| *Program and Shelter Accommodations and Adaptations* (α = 0.688) | | |
| provides dietary accommodations for refugee families in shelter. | 1.22 | 0.56 |
| offers multiple religious accommodations for refugee families in shelter. | 1.3 | 0.594 |
| has established partnerships with legal service providers who have experience in working with the refugee community. | 1.15 | 0.461 |
| has established partnerships with health care providers who have experience in working with the refugee community. | 1.33 | 0.663 |
| partners with or employs mental health providers who are trained in trauma-informed care with refugee populations. | 1.27 | 0.61 |
| *Agency Approaches to Language Concerns* (α = 0.714) | | |
| provides translated hard copy materials in an array of refugee languages. | 2.71 | 0.922 |
| staff take measures to verbally convey necessary content to them in their native language when working with a client who is illiterate in their native language, | 3.42 | 0.919 |
| uses adult family members of refugee clients for translation purposes. | 2 | 1.011 |
| uses friends of refugee clients for translation purposes. | 1.91 | 0.905 |
| uses the children of refugee clients for translation purposes. | 1.71 | 0.922 |
| offers multiple refugee language options on our website. | 1.68 | 1.105 |
| provides a multilingual library for families who are in shelter. | 1.8 | 1.036 |
| uses paid staff for translation purposes when working with refugee clients. | 3.04 | 1.103 |
| uses regular volunteers for translation purposes when working with refugee clients. | 2 | 1.043 |

Notes: Respondents rated each item on a scale from 1 = never to 4 = often.

*3.4. Data Analysis*

Given the exploratory nature of this study, the data analysis presented here is limited to descriptive and bivariate analyses. Frequency distributions and descriptive statistics are reported visually and descriptively below. In addition, correlation analyses that examined the relationship between agency characteristics and the aforementioned six domains of cultural responsiveness are reported.

## 4. Results

*4.1. Agency Familiarity with Refugee Communities*

As noted above, the median number of refugees served annually among responding agencies was 20. A full 44.2% of respondents reported serving 10 or less refugees per year. It is important to note again that the sampling frame for agencies was 212 resettlement cities across the U.S., where refugee communities comprise a nontrivial proportion of the wider population, sometimes upwards of 50,000. Agency directors were asked to identify the top three refugee groups (country of origin) in their city from an exhaustive dropdown list. Respondent choices were then compared to the Refugee Processing Center data for their state's resettlement cities over the past 5 years.[1] Just over half of respondents (53%) did not correctly identify the country of origin of any of the top three groups in their area, and the remaining identified one (38%) and two (9%) groups correctly. No respondents correctly identified the top three groups according to the Refugee Processing Center data.

*4.2. Culturally Responsive Organizational Practices*

A series of Likert scale questions asked survey respondents to rate the frequency with which various practices existed within their agency. Each item ranged on a scale from 1 = never to 4 = often, with higher mean scores indicating higher frequency for which the practices occur. When asked about agency values and practices, respondents indicated that their agency's staff were frequently knowledgeable about the cultural practices, values, and norms of refugee communities, but the practice of actively developing domestic violence leaders and advocates from within refugee communities was less common. Concerning agency staffing and programmatic practices, agencies most frequently responded that they promoted, mentored, and recruited bicultural and bilingual staff. Less often, however, did agencies report that they held multicultural events to promote cultural immersion or hire staff specifically from the refugee community to help with engagement and outreach. In fact, 2 out of 3 respondents (67%) reported they seldom or never hired from the refugee community to help with engagement and outreach. In terms of interagency collaboration, agencies reported that they more frequently worked to build relationships with other domestic violence service providers, as well as with refugee services offices, but they less often partnered with medical providers and faith leaders from refugee communities.

When asked about agency systems and processes for assessment, agencies reported more frequently assessing cultural competence practices, but less frequently reported that they use culturally relevant screening and assessment tools. When it came to accommodations and adaptations they make in their programs and shelter, the most frequent agency practice was establishing partnerships with legal service providers who have experience in working with the refugee community. In terms of addressing language concerns associated with working with the refugee community, agencies most often took measures to verbally convey necessary content to clients in their native language. Utilizing paid staff for translation purposes as well as providing translated hardcopy materials in accessible languages were other frequent practices among agencies. Agencies were less likely to provide multilingual libraries for clients in shelter or offer multiple refugee language options on agency websites.

---

1   http://www.wrapsnet.org/admissions-and-arrivals.

Among all six domains of cultural responsiveness, agencies reported most frequently enacting culturally responsive agency staffing and programmatic practices. Conversely, agencies less often reported enacting program and shelter accommodations and adaptations. All individual items within each of the six domains can be seen in Table 2.

### 4.3. Language Concerns

In addition, the respondents were asked to report whether or not their agency uses telephone translation services, and the overwhelming majority of agencies do (83.7%). However, only 47% of respondents could name the translation service their organization uses, and only 37% indicated that their agency dedicated a portion of their annual operating budget toward translation services.

### 4.4. Bivariate Results

Correlation analyses were conducted to examine the extent to which agency characteristics were associated with various domains of agency cultural responsiveness. As described in the instrumentation section and presented above in Table 2, domains of cultural responsiveness were calculated as composite scores and thus, reflect all items within each category. A number of agency characteristics were associated with one or more of the six domains of agency cultural responsiveness. The number of years the agency has been operating was negatively correlated to agency approaches to language concerns and agency systems/processes for assessment. In other words, the longer an agency had been in operation, the less often they reported implementing culturally responsive practices in language and assessment. The number of paid staff and number of clients served annually at an agency were not associated with any of the six domains of cultural responsiveness, but the number of volunteers was associated positively with program/shelter accommodations and adaptations. The annual budget was positively associated with agency values and practices, but the annual translation budget was not associated with any of the six domains of cultural responsiveness. One agency characteristic emerged as a key characteristic associated with multiple domains of cultural responsiveness: the percentage of total clients who are refugees within an annual period was positively associated with agency values and practices, agency staffing and programmatic practices, and interagency collaboration and outreach. However, the strength of these correlations only ranges from 0.296 to 0.347. Nearly every cultural competence domain was highly correlated with each other. Please see Table 3 for additional details.

**Table 3.** Pearson Correlations among Agency Characteristics and Domains of Agency Cultural Responsiveness.

| | Domains of Agency Cultural Responsiveness | 1 | 2 | 3 | 4 | 5 | 6 | 7 | 8 | 9 | 10 | 11 | 12 | 13 |
|---|---|---|---|---|---|---|---|---|---|---|---|---|---|---|
| 1 | Years Agency Operating | - | | | | | | | | | | | | |
| 2 | Number of Paid Staff | 0.298 * | - | | | | | | | | | | | |
| 3 | Number of Volunteers | 0.417 ** | 0.291 * | - | | | | | | | | | | |
| 4 | Number of Clients Served Annually | 0.175 | 0.748 ** | 0.173 | - | | | | | | | | | |
| 5 | Annual Budget | 0.357 * | 0.962 ** | 0.557 ** | 0.414 ** | - | | | | | | | | |
| 6 | Annual Translation Budget | 0.255 | 0.874 ** | 0.099 | 0.787 ** | 0.583 ** | - | | | | | | | |
| 7 | Percentage of Total Clients Who Are Refugees | −0.208 | −0.117 | −0.097 | −0.088 | −0.042 | −0.137 | - | | | | | | |
| 8 | Agency Values and Practices | −0.17 | −0.142 | −0.278 | 0.054 | −0.317 * | 0.069 | 0.316 * | - | | | | | |
| 9 | Agency Staffing and Programmatic Practices | −0.195 | −0.052 | −0.229 | 0.117 | −0.226 | 0.124 | 0.347 * | 0.841 ** | - | | | | |
| 10 | Interagency Collaboration and Outreach | −0.134 | −0.066 | −0.125 | 0.07 | −0.194 | 0.015 | 0.296 * | 0.844 ** | 0.698 ** | - | | | |
| 11 | Agency Systems and Processes for Assessment | −0.297 * | 0.027 | −0.24 | 0.216 | −0.12 | 0.269 | 0.275 | 0.762 ** | 0.820 ** | 0.671 ** | - | | |
| 12 | Program and Shelter Accommodations and Adaptations | −0.026 | 0.004 | 0.321 * | −0.096 | 0.237 | −0.196 | −0.177 | −0.406 ** | −0.304 * | −0.323 * | −0.338 * | - | |
| 13 | Agency Approaches to Language Concerns | −0.332 * | −0.1 | −0.26 | −0.013 | −0.206 | 0.1 | 0.244 | 0.653 ** | 0.747 ** | 0.632 ** | 0.736 ** | −0.268 | - |

Note: * indicates statistical significance at the $p < 0.05$ level; ** indicates statistical significance at the $p < 0.01$ level.

## 5. Discussion

The study results indicate there is a gap between domestic violence agencies and refugee communities in terms of familiarity. The results showed that by and large, executive directors were not able to name the most predominant refugee groups in their city. This is meaningful because the DV agencies surveyed were located in refugee resettlement cities, where refugee populations comprise a nontrivial portion of the wider population. When agencies lack even the basic knowledge of what refugee communities are most predominant in their city, they most likely lack more specific knowledge regarding cultural values and considerations within these communities that would improve their ability to effectively meet the needs of refugee DV survivors. For example, if an organization knew that one of the predominant refugee communities in their city was Somali, they might develop a basic understanding of Sunni Islamic traditions which could guide organizational outreach efforts. This finding connects to the number of refugee clients served per year—where nearly half of reporting agencies indicated serving 10 or fewer refugees annually. Without enough exposure, agencies may not experience a sense of urgency to adapt their services to better serve refugee communities. However, it should be reiterated that the responding agencies were located in resettlement cities where overall refugee populations comprise a meaningful proportion of the wider population. Lack of agency exposure to refugee communities should not be equated with a lack of need within refugee communities

In line with a general lack of familiarity with the refugee communities in their city, executive directors generally reported lower frequencies of agency outreach efforts to more informal refugee community contacts like faith leaders. The results suggested that, in general, DV agencies could improve interagency collaboration with refugee service providers as well as meaningful engagement with local refugee community leaders. In their 2009 report for the Family Violence Prevention Fund, Runner, Yoshihama, and Novick discuss how some DV organizations report little success in engaging existing community leaders in refugee and immigrant communities in the fight against domestic violence. Some of this may be due to a lack of organizational resources, and our results suggest this may be due to a lack of organizational prioritization.

Although DV agencies reported less frequently hiring from refugee communities, they reported more frequently hiring bilingual staff and encouraging cultural competence training. Agencies generally reported frequently engaging in organizational practices that promote dialogue and critical introspection among their staff, but less frequently reported efforts that promoted bidirectional exchanges with refugee communities. There seemed to be an inherent disconnect between how often agencies reported that they implement culturally responsive language practices for refugee communities and their agency budget for translation. While more than 8 out of 10 agencies reported that they use a translation service, almost 2 out of 3 agencies reported that they did not have an annual budget to support translation services. Agencies reported that they more often use formal translation services for help with refugee clients, as opposed to friends, family, and children; however, it is unclear how these services are being supported in agency budgets.

It is important to note that almost half (44%) of the agencies reported that 5% or more of their annual client population were from refugee communities in their city. When more than 5% of an agency's client base share an important identifying characteristic such as refugee status, it is critical that the agency develop and implement multi-level organizational approaches that support cultural responsiveness. Although among some of the weaker correlations, analyses showed that the percentage of clients served in the previous year who were refugees was positively associated with three of the six domains of cultural responsiveness, indicating that ongoing contact and cultural exchange can catapult organizations toward a deeper practice of cultural responsiveness.

Counterintuitively, the resources agencies allocate for translation services (one of the most critical supports according to the literature) was not associated with any of the domains of cultural responsiveness. Further, the length of time the agency had operated was inversely correlated with systems and processes for assessment and approaches to language concerns. This could reflect an entrenchment in prior ways of operating among older organizations. Older DV organizations are often

better resourced financially and stand to make the biggest impact in their cities. However, if these same organizations are not incorporating culturally responsive organizational practices, refugee survivors will continue to pay the price. No other organizational structure variables were correlated with agency cultural responsiveness domains.

The associations between cultural responsiveness domains were very strong and predominantly positive (with the exception of program/shelter accommodations/adaptations). From agency values/practices to agency approaches to language concerns, the correlations among these domains ranged from 0.63 to 84. Specifically, the correlations between language concerns and other domains of cultural responsiveness were high. When agencies are actively taking steps to translate materials, provide translation services, and generally communicate with clients in respectful ways that prioritize their lived experience, it is not surprising that agency values and other domains of cultural responsiveness would be thriving. These findings may point to the idea that when agencies enact culturally responsive practices in one domain, they are more likely to enact them in another. These practices can build on each other and lead to further improvements. For example, when an agency commits to hiring bilingual and bicultural staff, the agency perspective will broaden and new areas for improvement will be identified and rectified. These practices can domino into a more culturally responsive system on the whole as each incremental improvement spurs the agency to take action in the next. Findings from the bivariate analyses underscore a resounding theme in the literature: that agencies must take *active* steps to build a more culturally responsive organization.

*Limitations and Future Directions*

There are a number of limitations associated with this study. The first and foremost is the likelihood that agencies overestimated their cultural responsiveness due to social desirability and thus, the results may not accurately reflect the extent to which cultural competence and responsiveness permeates their agency. Secondly, the response rate of the survey (28%) likely points to selection bias and could further skew the results to overrepresenting the levels of cultural competence and responsiveness among domestic violence organizations. It is possible that organizations already invested in this work self-selected into the survey. Organizations that were more uncertain about their ability to serve refugee populations are more likely to have opted out of the survey, as Daniels and Belton (2015) found. Thus, the findings of this study cannot be generalized to all DV agencies in U.S. resettlement cities. Finally, due to the study's descriptive and exploratory nature, the results do not link culturally responsive strategies to specific outcomes for victims of domestic violence. The results give us a starting point for understanding the extent of cultural competence and responsiveness among domestic violence organizations; however, they do not offer clear guidance for how to prioritize organizational change to effect positive outcomes for refugee clients. This then, is a natural next for future research. Evaluating the efficacy of specific culturally competent practices and organizational policies is critical in order to inform evidence-based practice with refugee domestic violence survivors.

**Author Contributions:** Conceptualization, J.L.L. and K.M.S.; methodology, J.L.L. and K.M.S.; formal analysis, J.L.L. writing—original draft preparation, J.L.L., K.M.S., and T.H.; writing—review and editing, J.L.L.; funding acquisition, J.L.L. and K.M.S. All authors have read and agreed to the published version of the manuscript.

**Funding:** This work was funded primarily by Fight Against Domestic Violence, with additional support from the Mountain West Center for Regional Studies and the College of Humanities and Social Sciences at Utah State University. The views presented in this paper do not necessarily reflect the views of the funders.

**Acknowledgments:** The authors would like to thank Anna Z. Blau and Laura M. Mora from the International Women's House for their expert consultation on instrumentation development. In addition the authors would like to thank the students from the 2018 graduating class of Utah State University's Master of Social Work program for their assistance with the sampling strategy.

**Conflicts of Interest:** The authors declare no conflict of interest.

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
