# Peer review of "Assessing Organizational Cultural Responsiveness among Refugee-Servicing Domestic Violence Agencies"

_socsci, doi:10.3390/socsci9100176_

Round 1
Reviewer 1 Report
Review of 903069
This paper reports a early, small-scale study in a contemporary area of social interest. The paper addresses agency responsiveness to the important social problem of domestic violence for refugees in the USA. The extent and urgency of society’s responses to domestic violence warrants publishing early work like this in order to bring attention to the issues and to foster further research studies. The difficulties of refugees compounds the domestic violence issue for people who have already be subjected to violence in many forms.
What this paper can contribute is a small correlational study of the perceptions of directors of 70 organisations who have what appears to be minimal but significant interactions with clients from refugee communities who experience domestic violence. The study reports some patterns of interaction with refugee clients, and organisational perceptions and emphases. I think the authors could revise their paper with a useful, more modest focus, and contribute to a small body of research about an important social issue. Accordingly, I recommend that the authors do some careful revision on their presentation in order to make it more suitable for an international journal. My comments are for a revision and I would welcome a revised paper. Comments are aimed at assisting the authors to make their revisions.
- The focus and aims of the paper
There are two descriptions of the aims of the study that I do not see as being coordinated (on p2., and p. 4,5). At p2, l50ff, the authors refer to the need to “effectively serve refugee survivors”, and to ‘assess current best practice’ I am not sure how well those concepts of “effectively serve” and “best practice” are reflected in the design and presentation of the study. There is an argument in the Introduction that I do not think fits the Results. For instance, ‘effectiveness’ seems to be beyond the scope of reporting patterns of service details and interactions with refugee clients. I think the reporting (not assessment or evaluation) of organisational perceptions is a more appropriate description of the study and its contribution. I cannot see where “best practice” fits the data.
Then p4,5 Section 2.4 gives 2 definite aims for the study (1) explore the extent to 197 which domestic violence agencies in resettlement cities across the United States are culturally competent; and (2) identify the characteristics of culturally responsive agencies (p. 5, l.198ff). Again I am not sure how well the scales (and their items) reflect ‘cultural competence’ although I think the authors may want to mount an argument for that. I think the paper would be better served by a more modest composite descriptor of the scales as “approaches” or “operational perceptions” or something less grand and closer to the data would be useful. The authors then could draw out some inferences about these and the patterns – and argue for their status as indicators of ‘cultural competence’ in interacting with refugees in the Discussion and conclusions. I think the emphasis on patterns of activity “.. are enacting practices and policies that are culturally responsive to refugee survivors” with a specific focus on familiarity, responsiveness, values etc. (p.13, l 379 ffing) would be a useful motif for the Introduction.
I find the focus on the individual characteristics and educational background of the directors a distraction away from the activities of the agencies. I am concerned about the combination of agency values, activities and approaches and the detailed reporting of director’s characteristics in the Method (p5. 3.2). If this is considered important, then the authors need to make more of it and build it into the introductory argument. I would prefer to keep the focus on the organisational rather than on personal characteristics of directors. If individual profiles of directors are considered an important part of the study, then perhaps the focus should have been on directors and their approaches rather than using their perceptions as indicators of organisational profiles. The authors should clear up this combination and make their sampling focus clearer and relate it to the data analysis. Tables 2 to 9 focus on the agencies and that seems to be the actual perspective – and organisational responsiveness.
- The Argument
There is a disjunction between the Introduction and its argument about responsiveness to domestic violence among refugees and specifics of the Method and Results. The argument in the Discussion also ventures too far away from the results. Specifically I think the Section 5.1 Practical recommendations (p.13) ventures too far from the scope of the study.
The integrity of the paper and the actual argument of it as a total paper need to be refined and reshaped, and refocused. The Introduction is well written and builds a nice case for the importance of organisational responsiveness to the domestic violence experiences in refugee communities in major geographical centers in the USA and management of organisational practice for refugee clients. However, the argument and background do not lead into the actual design and presentation of the study.
- Data analysis and Results Presented
The analysis should be directly focused on the correlational analysis (Table 10). The authors report 6 scales with reasonable to good internal consistency (3.3). I think these scales should be the focus on the analyses and especially the Results Section. These also could be correlated with agency trends (numbers of clients etc). The authors hint at the usefulness of a correlational study (p. 7, l. 266) but then present 8 tables of individual items. Even as a preliminary study, this representation of individual items is unbalanced and not as informative as a small-scale correlational study of organisational patterns. The psychometric properties and descriptive usefulness of the scales would serve a stronger approach.
Psychometrically, I think the scales would be more compelling as reports of frequency if ‘never’ was ‘0’ instead of ‘1’, but can be a point of disagreement.
- Tables and Figures
Ten tables are rarely accepted in a published paper, and certainly not here. I would see one or perhaps one on agency profiles and demographics and a correlation matrix as warranted. I do not think that Figure One adds to the value of the article. I think the tables could be reduced to one comprehensive, cleaned up Table (10 at present). Some of the organisational demographics also could be added to the correlational analysis (see Table 1).
In general, I think the topic is timely, the sample is small and not representative of the field – so the authors need to make a case for the usefulness of the study. I recommend a careful revision and revisiting of the links between the sections, a refocus and rewrite and presentation as a modest correlational study.
Author Response
Many thanks to the reviewer for thoughtful, constructive feedback that has allowed us to improve our paper. Below we have organized our point by point response to the reviewer in bolded text. Revisions in the manuscript itself can be seen in highlighted yellow.
Reviewer 1
This paper reports a early, small-scale study in a contemporary area of social interest. The paper addresses agency responsiveness to the important social problem of domestic violence for refugees in the USA. The extent and urgency of society’s responses to domestic violence warrants publishing early work like this in order to bring attention to the issues and to foster further research studies. The difficulties of refugees compounds the domestic violence issue for people who have already be subjected to violence in many forms.
What this paper can contribute is a small correlational study of the perceptions of directors of 70 organisations who have what appears to be minimal but significant interactions with clients from refugee communities who experience domestic violence. The study reports some patterns of interaction with refugee clients, and organisational perceptions and emphases. I think the authors could revise their paper with a useful, more modest focus, and contribute to a small body of research about an important social issue. Accordingly, I recommend that the authors do some careful revision on their presentation in order to make it more suitable for an international journal. My comments are for a revision and I would welcome a revised paper. Comments are aimed at assisting the authors to make their revisions.
The focus and aims of the paper
There are two descriptions of the aims of the study that I do not see as being coordinated (on p2., and p. 4,5). At p2, l50ff, the authors refer to the need to “effectively serve refugee survivors”, and to ‘assess current best practice’ I am not sure how well those concepts of “effectively serve” and “best practice” are reflected in the design and presentation of the study. There is an argument in the Introduction that I do not think fits the Results. For instance, ‘effectiveness’ seems to be beyond the scope of reporting patterns of service details and interactions with refugee clients. I think the reporting (not assessment or evaluation) of organisational perceptions is a more appropriate description of the study and its contribution. I cannot see where “best practice” fits the data.
Substantial changes have been made to the purpose and aims in the places the reviewer mentions. We believe the term “assessment” of organizational perceptions and practices is accurate, however. We have removed “best practices” from the text and instead have standardized language regarding assessing organizational perceptions and practices. We believe “assessment” is a fair term to use for a couple of reasons. One, Caldaza & Suarez-Balcazar’s (2014) model of cultural competence provides a useful tool that assesses cultural competence at the broadest level of an organization and how that influences cultural competence at the staff and program levels. We have adjusted language in the paper to accurately describe Caldaza & Suarez-Balcazar’s model as an assessment tool. Secondly, Caldaza & Suarez-Balcazar’s work is grounded in a much larger body of cultural competence assessment tools (i.e., Georgetown University’s National Center for Cultural Competence). For these reasons, referring to our study as an assessment of organizational cultural competence/responsiveness is accurate.
Then p4,5 Section 2.4 gives 2 definite aims for the study (1) explore the extent to 197 which domestic violence agencies in resettlement cities across the United States are culturally competent; and (2) identify the characteristics of culturally responsive agencies (p. 5, l.198ff). Again I am not sure how well the scales (and their items) reflect ‘cultural competence’ although I think the authors may want to mount an argument for that. I think the paper would be better served by a more modest composite descriptor of the scales as “approaches” or “operational perceptions” or something less grand and closer to the data would be useful. The authors then could draw out some inferences about these and the patterns – and argue for their status as indicators of ‘cultural competence’ in interacting with refugees in the Discussion and conclusions. I think the emphasis on patterns of activity “.. are enacting practices and policies that are culturally responsive to refugee survivors” with a specific focus on familiarity, responsiveness, values etc. (p.13, l 379 ffing) would be a useful motif for the Introduction.
We have added significant background on how the items were developed through Caldaza & Suarez-Balcazar’s work and the decades-long exploration of these concepts in nursing, psychology, and social work. We hope it is clear to the reader that these items were not generated out of thin air, but rather that they map on to key theoretical concepts presented in the literature, and specifically delineated by Caldaza & Suarez-Balcazar, two leaders in the field of cultural competence. We believe using the term “practices” and “domains” when discussing individual items and composite scores is not an overstatement given the revisions made to the manuscript throughout to clarify how these items fit into the larger picture of the study of cultural competence.
I find the focus on the individual characteristics and educational background of the directors a distraction away from the activities of the agencies. I am concerned about the combination of agency values, activities and approaches and the detailed reporting of director’s characteristics in the Method (p5. 3.2). If this is considered important, then the authors need to make more of it and build it into the introductory argument. I would prefer to keep the focus on the organisational rather than on personal characteristics of directors. If individual profiles of directors are considered an important part of the study, then perhaps the focus should have been on directors and their approaches rather than using their perceptions as indicators of organisational profiles. The authors should clear up this combination and make their sampling focus clearer and relate it to the data analysis. Tables 2 to 9 focus on the agencies and that seems to be the actual perspective – and organisational responsiveness.
Thank you for this recommendation. We agree that reporting directors demographic information is a departure from the aim of the study and have removed this table and text.
The Argument
There is a disjunction between the Introduction and its argument about responsiveness to domestic violence among refugees and specifics of the Method and Results. The argument in the Discussion also ventures too far away from the results. Specifically I think the Section 5.1 Practical recommendations (p.13) ventures too far from the scope of the study.
We have made substantial changes to the background, results, and discussion in response to this comment. In addition, we removed 5.1 (Practical Recommendations) from the paper and agree that it was a bit off course. It will be better used in a practice brief or another practitioner-focused work.
The integrity of the paper and the actual argument of it as a total paper need to be refined and reshaped, and refocused. The Introduction is well written and builds a nice case for the importance of organisational responsiveness to the domestic violence experiences in refugee communities in major geographical centers in the USA and management of organisational practice for refugee clients. However, the argument and background do not lead into the actual design and presentation of the study.
We have made significant revisions to the paper thanks to your helpful feedback. These can be seen highlighted in yellow throughout. Key deletions were made as well. We believe the entire paper flows better now and is more cohesive.
Data analysis and Results Presented
The analysis should be directly focused on the correlational analysis (Table 10). The authors report 6 scales with reasonable to good internal consistency (3.3). I think these scales should be the focus on the analyses and especially the Results Section. These also could be correlated with agency trends (numbers of clients etc). The authors hint at the usefulness of a correlational study (p. 7, l. 266) but then present 8 tables of individual items. Even as a preliminary study, this representation of individual items is unbalanced and not as informative as a small-scale correlational study of organisational patterns. The psychometric properties and descriptive usefulness of the scales would serve a stronger approach.
With the revisions to the number of tables, we hope that the correlation analyses stand out more as the final table in the paper. Having each of the individual items in table 2 is helpful for the reader to understand the items that comprised the scales. Particularly because we hope for this paper to have a wide readership, especially including practitioners who are interested in comparing how they measure up to other DV agencies who are serving refugees. Without the lengthy list of individual items, we cannot serve this practical purpose. Additionally, we included reliability information in table 2 as a reminder to the reader prior to the section on the correlation analyses. Finally, we inserted more discussion of the correlation analyses in the discussion section.
Psychometrically, I think the scales would be more compelling as reports of frequency if ‘never’ was ‘0’ instead of ‘1’, but can be a point of disagreement.
Fair point. If we were doing something more statistically sophisticated with these scales, we would recode as you suggest. As we are simply reporting descriptives in this paper, we have kept them as is.
Tables and Figures
Ten tables are rarely accepted in a published paper, and certainly not here. I would see one or perhaps one on agency profiles and demographics and a correlation matrix as warranted. I do not think that Figure One adds to the value of the article. I think the tables could be reduced to one comprehensive, cleaned up Table (10 at present). Some of the organisational demographics also could be added to the correlational analysis (see Table 1).
Thank you for your advice on this. We have reduced the number of tables to 3 (agency characteristics, correlation matrix, and organizational perceptions/practices. We believe it reads with more clarity and focus.
In general, I think the topic is timely, the sample is small and not representative of the field – so the authors need to make a case for the usefulness of the study. I recommend a careful revision and revisiting of the links between the sections, a refocus and rewrite and presentation as a modest correlational study.
Yes, we agree that the sample size is small; however, we want to reiterate that the sampling frame was DV agencies with shelters in resettlement cities in the U.S. There are 212 resettlement cities and a total of 301 DV agencies within these cities were identified through the National Coalition Against Domestic Violence’s DV shelter search tool, ensuring complete coverage. Although we only had 70 valid survey responses, out of the total sampling frame (301), this is not a trivial amount. We have ensured that discussion about selection bias and representativeness is clear in the limitations section. However, given that there are currently no data on this topic outside of qualitative interviews with agencies (Runner, Novick, & Yoshihama, 2009, we agree that the topic is timely and we believe it will provide a critical launch point for future studies in this area.
Reviewer 2 Report
he manuscript tends to describe a relevant issue that has implications in social responses to immigrant/refugee populations and their needs. The general area upon which this manuscript touches, cultural responsive social service- services for refugee’ victims of domestic violence is timely and necessary considering the public discourse about migration, treatment of immigrant population and their access to resources once residing in the U.S. Line 8: It should be explained what the author views as the difference between refugees and other immigrants. The number of undocumented migrants is increasing, and the needs of this population are not very different: they often ran from violence in their home country, suffer the cumulative trauma from their journey, face cultural challenges in the host country and lack access to resources as a result of their undocumented status etc. The terms “migrant” and “refugee” are often used interchangeably but it is important to distinguish between them as there is a legal difference. In the introduction section, it would be beneficial to explain what the resettlement cities are, what is the approximate number of refugee populations in such cities, what countries the refugee are coming from etc. Line 277 278 should be supported by the introduction about the resettlement cities. Line 274 Section 4.1. It would be interesting if the authors explain the differences between the country of origin and the assumption about the country of origin by the research participants. The lack of knowledge about the country of origin has ab impact on social services as the specific cultural attitudes, values and social structures within the refugee communities are not recognized and appropriately addressed.Author Response
Thank you for your helpful comments as we worked to improve, clarify, and streamline our paper. We appreciate your thoughtful recommendations. Our responses can be seen in bold below. In the manuscript you'll see revisions highlighted in yellow.
Reviewer 2
The manuscript tends to describe a relevant issue that has implications in social responses to immigrant/refugee populations and their needs. The general area upon which this manuscript touches, cultural responsive social service- services for refugee’ victims of domestic violence is timely and necessary considering the public discourse about migration, treatment of immigrant population and their access to resources once residing in the U.S.
Line 8: It should be explained what the author views as the difference between refugees and other immigrants. The number of undocumented migrants is increasing, and the needs of this population are not very different: they often ran from violence in their home country, suffer the cumulative trauma from their journey, face cultural challenges in the host country and lack access to resources as a result of their undocumented status etc. The terms “migrant” and “refugee” are often used interchangeably but it is important to distinguish between them as there is a legal difference.
Thank you for this comment. We have inserted the legal definition of refugee in the manuscript and added a bit of context to the issues you raised above. We agree that there are significant similarities, but also key differences regarding legal status.
In the introduction section, it would be beneficial to explain what the resettlement cities are, what is the approximate number of refugee populations in such cities, what countries the refugee are coming from etc.
We have added some further detail regarding resettlement cities, numbers, and from what primary countries of origin in the last few years. There are 212 resettlement cities across 49 U.S. states – some states have as many as 13 resettlement cities but most have just 1.
Line 277 278 should be supported by the introduction about the resettlement cities.
We added the total number of resettlement cities to this line, reinforcing the discussion in the sampling section.
Line 274 Section 4.1. It would be interesting if the authors explain the differences between the country of origin and the assumption about the country of origin by the research participants. The lack of knowledge about the country of origin has ab impact on social services as the specific cultural attitudes, values and social structures within the refugee communities are not recognized and appropriately addressed.
We agree that this is an interesting finding that has significant bearing on how organizations interact with refugee communities in their city. We have added more discussion of this finding in the
Round 2
Reviewer 1 Report
Review of Resubmit of 903069
The authors have made good work of tightening and refining the paper. I think the 3 tables work much better, and that it is valuable to add the scale alphas to Table 2. I think the paper now can make a useful contribution to an area where there is little evidence of services for refugees in the USA. I have a few more comments that will strengthen the presentation and especially the Discussion of the correlations. Another round of refinement is called-for and will be worthwhile. Good wishes for a final editing.
- I think it would be useful to remove the one mean stat from Table 1 and report the mean and s.d. in the text. That would leave Table 1 uniformly reporting medians.
- Section 4.4 is an important part of the study. I don’t think you need “As described … each category” (l307). The description of the scales could be left in the Method – say in section 3.3 (around l244). The substantive reporting is part of the Results Section.
- What follows in 4.4 could be refined and focused on the correlations. I think the correlations are under-worked a little. I would expect a little more reading-off of the relative size of correlations and some lead-in or summary comment that that was little association between the organizational structures (e.g., budget, numbers) and activities and values. I think the values of some pairs could be well reproduced in the text to make a strong link between text and table. There are some very high correlations (e.g., values and staffing practices with assessment processes). Do you want to highlight those? Are the variables strong enough to carry that weight? If so – emphasize them. I wonder if ‘percentages of total clients who are refugees’ is a reasonable variable. It doesn’t associate well with any activities. Do you need to say why? Would it be reasonable to back it up with total number of refugee clients?
In relation to this variable – I do not think the correlations support the claim about the percentage at l374ff in the Discussion. .347 is the largest correlation – not impressive at all. I think the whole interpretation of correlation size needs a rethink and rewrite. You have some very high correlations in the later variables – work on those. - The correlations with ‘language concerns’ are important for your focus. Could you comment briefly on them?
- In the Discussion – I would recommend omitting lies 335-337 that are fillers. Begin with ’Study results indicate’ (l337)
- ‘On the contrary..metric’ (l3690 makes no sense – remove that phrase. Begin the sentence with ‘nearly one-third…’ (l370)
Author Response
The authors have made good work of tightening and refining the paper. I think the 3 tables work much better, and that it is valuable to add the scale alphas to Table 2. I think the paper now can make a useful contribution to an area where there is little evidence of services for refugees in the USA. I have a few more comments that will strengthen the presentation and especially the Discussion of the correlations. Another round of refinement is called-for and will be worthwhile. Good wishes for a final editing.
Thank you for your comments on round one and for your comments here to further refine the paper. We have made insertions in the manuscript highlighted in green and referenced by line below.
I think it would be useful to remove the one mean stat from Table 1 and report the mean and s.d. in the text. That would leave Table 1 uniformly reporting medians.
We have made this correction.
Section 4.4 is an important part of the study. I don’t think you need “As described … each category” (l307). The description of the scales could be left in the Method – say in section 3.3 (around l244). The substantive reporting is part of the Results Section.
We removed this line in section 4.4 and agree that the bases are covered in section 3.3.
What follows in 4.4 could be refined and focused on the correlations. I think the correlations are under-worked a little. I would expect a little more reading-off of the relative size of correlations and some lead-in or summary comment that that was little association between the organizational structures (e.g., budget, numbers) and activities and values. I think the values of some pairs could be well reproduced in the text to make a strong link between text and table. There are some very high correlations (e.g., values and staffing practices with assessment processes). Do you want to highlight those? Are the variables strong enough to carry that weight? If so – emphasize them. I wonder if ‘percentages of total clients who are refugees’ is a reasonable variable. It doesn’t associate well with any activities. Do you need to say why? Would it be reasonable to back it up with total number of refugee clients?
We have reworked some of this discussion to better highlight major findings from this table. In particular, we’ve added some discussion about the high correlation between all of the cultural responsiveness composite variables. This may point to the idea that when an organization does better in one domain, it opens the organizational perspective to improve in other domains. We also made note that the program and shelter accommodations and adaptations construct was negatively correlated with other culturally responsive composite variables. This may be a reflection of the lower reliability (.688) for that composite variable. Additional discussion can be found on lines 389-404.
As for the question about percentage of refugees, on an earlier analysis we had used total number of refugees in addition to the percentage variable. The strength of the associations were only slightly higher for the percentage variable compared to the total number. However, we thought the percentage variable made more theoretical sense. For example, if an agency served 1,000 clients a year but only saw 20 refugee clients a year, they would be more likely to see this a fringe group than an agency who served 100 clients a year, 20 of whom were refugees. The percentage variable actually does correlate with agency values/practices; agency staffing/programmatic practices; and interagency collaboration and outreach, though the strength of the correlation statistics only range from .296 to .347. The discussion highlights these correlations on lines 378-391.
In relation to this variable – I do not think the correlations support the claim about the percentage at l374ff in the Discussion. .347 is the largest correlation – not impressive at all. I think the whole interpretation of correlation size needs a rethink and rewrite. You have some very high correlations in the later variables – work on those.
We added discussion on the stronger correlations as noted above and inserted a caveat in the results and discussion about the strength of correlations with the percentage of refugee clients variable.
The correlations with ‘language concerns’ are important for your focus. Could you comment briefly on them?
Yes, we commented on this on lines 394-398.
In the Discussion – I would recommend omitting lies 335-337 that are fillers. Begin with ’Study results indicate’ (l337)
These lines have been omitted.
‘On the contrary..metric’ (l3690 makes no sense – remove that phrase. Begin the sentence with ‘nearly one-third…’ (l370)
We deleted that entire sentence and added a bit more explanation about the total numbers of refugees served in paragraph one. This allows the later discussion to be more centered on the percentage of total clients variable and correlations.
